# MASS: MoErging through Adaptive Subspace Selection

**Donato Crisostomi***     **Alessandro Zirilli***

**Antonio Andrea Gargiulo**     **Maria Sofia Bucarelli**     **Simone Scardapane**

**Fabrizio Silvestri**     **Iacopo Masi**     **Emanuele Rodolà**

Sapienza University of Rome

`{crisostomi, zirilli, gargiulo, masi, rodola}@di.uniroma1.it`
`simone.scardapane@uniroma1.it`     `{bucarelli, fsilvestri}@diag.uniroma1.it`

## Abstract

Model merging has emerged as a lightweight alternative to ensembling, combining multiple fine-tuned models into a single set of parameters without additional training. However, existing methods rarely match the accuracy of individually fine-tuned models. We introduce MASS (MoErging through Adaptive Subspace Selection), a training-free approach that narrows this gap while maintaining near state-of-the-art performance across tasks. MASS leverages low-rank decompositions of task-specific updates, storing only the most salient singular components and merging them into a shared model. At inference, a data-free, non-parametric router selects the most relevant subspace (or combination of subspaces) based on intermediate features. This adds only a two-pass inference overhead and a $\sim 2\times$ storage cost relative to a single pretrained model, regardless of the number of tasks. Evaluated on CLIP-based image classification with `ViT-B-16`, `ViT-B-32`, and `ViT-L-14` across 8, 14, and 20 tasks, MASS achieves up to $\sim 98\%$ of the accuracy of separate fine-tuned models, establishing a new state-of-the-art while remaining far more storage-efficient than ensembling.

## 1 Introduction

Early deep learning models were trained from scratch, but the rise of large pretrained backbones shifted the focus to fine-tuning for specific tasks Devlin et al. [2019], Tan et al. [2018], Yosinski et al. [2014], Hu et al. [2022], Radford et al. [2021]. Today, the abundance of publicly available fine-tuned models[1] has sparked interest in *no-tuning* methods that exploit both the foundation model and existing fine-tuned endpoints.

Among these, *model merging* [Ilharco et al., 2023, Akiba et al., 2025, Yadav et al., 2023, Yu et al., 2024, Ainsworth et al., 2023, Crisostomi et al., 2025, Singh and Jaggi, 2020, Gargiulo et al., 2025, Zhou et al., 2024, Daheim et al.] offers a lightweight, storage-efficient alternative to ensembling by combining multiple fine-tuned models into a single parameter set. Early approaches such as `Task Arithmetic` [Ilharco et al., 2023] simply summed task vectors (fine-tuned minus pretrained weights), while later methods [Gargiulo et al., 2025, Daniel et al., 2025] preserved layer-wise structure for better accuracy. In particular, `Task Singular Vectors` (TSV) [Gargiulo et al., 2025] leveraged the low-rank structure of task updates, retaining most fine-tuned performance with only a few singular vectors per task.

However, current structured merging methods remain static: their aggregation weights do not adapt to the input, leading to suboptimal performance and significant accuracy drops for certain tasks.

---

[1]https://huggingface.co/docs/hub/models-the-hub

To address this, we introduce **MASS** (*Mo-Erging through Adaptive Subspace Selection*), which integrates the adaptivity of Mixture-of-Experts (MoE) [Shazeer et al., 2017, Eigen et al., 2014, Fedus et al., 2022, Du et al., 2022] with singular-vector-based merging. MASS dynamically routes inputs to the most relevant task subspaces (Fig. 1) without requiring task data or additional tuning. This is a key advantage in scenarios where only model checkpoints are available.

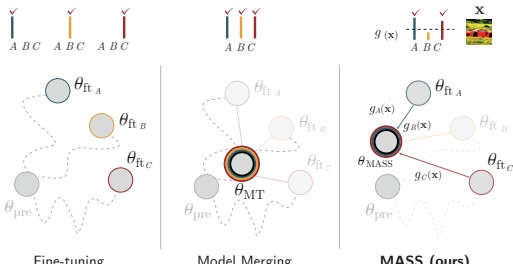

Figure 1: *(left)* Fine-tuning creates separate models for tasks A, B, and C. *(middle)* Model merging combines their task vectors $\{A, B, C\}$ using a fixed aggregation. *(right)* MASS stores $\theta_{\mathrm{pre}}$ and orthogonalized task singular vectors $V_\perp^\top$, and adaptively merges them at test time via a routing function $g(\mathbf{x})$ that selects task subspaces based on input features.

We evaluate MASS on `ViT-B-32`, `ViT-B-16`, and `ViT-L-14` backbones across 8, 14, and 20 tasks. Our method consistently outperforms existing merging techniques, recovering up to 95%–98% of the accuracy of individually fine-tuned models with only a modest overhead ($\sim 2\times$ inference and storage relative to the base model). In batched settings, MASS reduces accuracy loss to below 1% on most benchmarks.

Our key contributions are:

- Introducing MASS, a singular-vector-based merging method with adaptive input routing.
- Proposing a projection-based, data-free router that requires no additional fine-tuning.
- New state-of-the-art results in multitask merging across models and tasks.

## 2  Background

In this section, we introduce the key concepts underlying our approach.

**Task Vectors**  `Task Arithmetic` (TA) [Ilharco et al., 2023] represents each task as a *task vector*, i.e., the difference between fine-tuned and pretrained weights. A multitask model for $T$ tasks is obtained by summing these vectors:

$$\theta_{\mathrm{MT}} = \theta_{\mathrm{pre}} + \alpha \sum_{i=1}^{T} \tau_i\,, \tag{1}$$

where $\theta_{\mathrm{pre}}$ are the pretrained weights, $\alpha$ is a scaling factor, and $\tau_i = \theta_{\mathrm{ft}_i} - \theta_{\mathrm{pre}}$ is the task vector for task $i$. Following Gargiulo et al. [2025], we consider this operation layer-wise:

$$\theta_{\mathrm{MT}}^{(\ell)} = \theta_{\mathrm{pre}}^{(\ell)} + \alpha \sum_{i=1}^{T} \Delta_i^{(\ell)}, \tag{2}$$

where $\Delta_i^{(\ell)} = \theta_{\mathrm{ft}_i}^{(\ell)} - \theta_{\mathrm{pre}}^{(\ell)}$ is the task-specific weight difference for layer $\ell$. For matrix-shaped layers, $\Delta_i^{(\ell)}$ is referred to as the *per-layer task matrix*. For brevity, we omit the layer index.

**Task Singular Vectors**  Gargiulo et al. [2025] show that task matrices $\Delta_i$ exhibit strong low-rank structure. For each task $i$, they compute the SVD:

$$\Delta_i = U_i \Sigma_i V_i^\top\,,$$

and truncate it to rank $k$, keeping the top-$k$ singular vectors and values: $\tilde{U}_i$, $\tilde{V}_i$, and $\tilde{\Sigma}_i$. Stacking these across tasks yields $U = [\tilde{U}_1 \cdots \tilde{U}_T]$ (left TSVs), $V = [\tilde{V}_1 \cdots \tilde{V}_T]$ (right TSVs), and $\tilde{\Sigma}$ (block-diagonal with $\tilde{\Sigma}_i$). The multitask update is then expressed as:

$$\hat{\Delta} = U_\perp \Sigma V_\perp^\top\,, \tag{3}$$

where $U_\perp$ and $V_\perp^\top$ are orthogonalized to reduce inter-task interference. This effectively sums the top-$k$ rank-one updates per task while ensuring the task subspaces remain distinct (see Alg. 1 in the Appendix).

| | Method | ViT-B-32 | | | ViT-B-16 | | | ViT-L-14 | | |
|---|---|---|---|---|---|---|---|---|---|---|
| | | 8 tasks | 14 tasks | 20 tasks | 8 tasks | 14 tasks | 20 tasks | 8 tasks | 14 tasks | 20 tasks |
| Base | Zeroshot | $48.2_{(53.5)}$ | $57.2_{(63.6)}$ | $56.1_{(62.4)}$ | $55.3_{(59.3)}$ | $61.2_{(66.1)}$ | $59.7_{(64.5)}$ | $64.7_{(68.0)}$ | $68.2_{(72.1)}$ | $65.2_{(68.9)}$ |
| | Finetuned | $92.8_{(1.00)}$ | $90.8_{(1.00)}$ | $91.3_{(1.00)}$ | $94.6_{(1.00)}$ | $92.7_{(1.00)}$ | $93.1_{(1.00)}$ | $95.8_{(1.00)}$ | $94.2_{(1.00)}$ | $94.7_{(1.00)}$ |
| Fixed | Weight Averaging | $66.3_{(72.1)}$ | $64.3_{(71.1)}$ | $61.0_{(67.5)}$ | $72.2_{(76.6)}$ | $69.4_{(74.8)}$ | $65.3_{(70.3)}$ | $79.5_{(83.1)}$ | $76.7_{(81.1)}$ | $71.6_{(75.6)}$ |
| | Task Arithmetic Ilharco et al. [2023] | $70.7_{(76.5)}$ | $65.3_{(72.0)}$ | $60.5_{(66.7)}$ | $75.4_{(79.5)}$ | $70.5_{(75.8)}$ | $65.7_{(70.7)}$ | $84.9_{(88.6)}$ | $79.4_{(83.9)}$ | $74.0_{(78.0)}$ |
| | Consensus TA Wang et al. [2024] | $75.0_{(80.8)}$ | $70.3_{(77.3)}$ | $65.4_{(71.9)}$ | $79.3_{(83.8)}$ | $74.3_{(79.9)}$ | $69.7_{(74.9)}$ | $86.3_{(90.0)}$ | $82.2_{(86.9)}$ | $79.0_{(83.2)}$ |
| | TSV-M Gargiulo et al. [2025] | $85.8_{(92.3)}$ | $80.0_{(87.8)}$ | $77.0_{(84.2)}$ | $89.0_{(93.9)}$ | $84.5_{(91.0)}$ | $80.5_{(86.4)}$ | $92.9_{(96.9)}$ | $89.1_{(94.4)}$ | $87.7_{(92.5)}$ |
| | Iso-C Daniel et al. [2025] | $86.3_{(92.9)}$ | $80.3_{(88.1)}$ | $75.5_{(82.5)}$ | $90.6_{(95.6)}$ | $84.8_{(91.1)}$ | $79.6_{(85.4)}$ | $94.2_{(98.3)}$ | $89.3_{(94.5)}$ | $87.6_{(92.2)}$ |
| | Iso-CTS Daniel et al. [2025] | $86.2_{(92.8)}$ | $81.7_{(89.7)}$ | $78.1_{(85.5)}$ | $91.1_{(96.1)}$ | $86.4_{(92.8)}$ | $82.4_{(88.4)}$ | $94.7_{(98.8)}$ | $91.0_{(96.3)}$ | $90.1_{(94.9)}$ |
| MoE | **MASS** | $\mathbf{90.6_{(97.6)}}$ | $\mathbf{86.8_{(95.5)}}$ | $\mathbf{84.4_{(92.5)}}$ | $\mathbf{93.2_{(98.5)}}$ | $\mathbf{90.2_{(97.3)}}$ | $\mathbf{85.3_{(91.9)}}$ | $\mathbf{94.6_{(98.7)}}$ | $\mathbf{91.4_{(97.0)}}$ | $\mathbf{90.6_{(95.7)}}$ |

Table 1: Average absolute accuracy results on model merging benchmarks; subscript (in parentheses) is the normalized average accuracy.

## 3 Approach

Our approach consists of a one-time *fixed merging* step and an *adaptive inference* step.

**Fixed merging.** We first merge task-specific updates using TSV-M Gargiulo et al. [2025] to produce an encoder $\theta_{MT}$ capable of separating task subspaces. This step is input-independent and performed only once.

**Adaptive inference.** At test time, MASS dynamically routes each input $\mathbf{x}$ through four steps:

   (i) **First pass**: forward $\mathbf{x}$ through $\theta_{MT}$;
  (ii) **Routing**: compute the projection residual of intermediate activations onto each task subspace and select the lowest-residual tasks;
 (iii) **Adaptive merge**: combine the selected subspaces into $\Delta_{ada}$;
 (iv) **Second pass**: predict with $\theta_{pre} + \alpha\Delta_{ada}$.

**Projection-based routing.** Unlike existing routers, which require task data or additional training, our router is entirely data-free. For activations $\mathbf{z}_\ell$ at a chosen layer, we compute:

$$r_i = \left\| \mathbf{z}_\ell - V_i V_i^\top \mathbf{z}_\ell \right\|_2, \tag{4}$$

where $V_i$ is the matrix of right singular vectors for task $i$. Tasks with residuals below a threshold $\eta$ are selected. To prevent redundant directions from dominating, we discard highly similar subspaces during fixed merging using a cosine-similarity filter.

**Adaptive merging and prediction.** The selected subspaces $\Omega$ are merged via TSV-M to obtain $\theta_{MASS}$. Each corresponding task head $h_i$ produces logits $\mathbf{z}_i$, and the head with the highest confidence determines the final prediction:

$$(i^\star, c^\star) = \underset{(i,c) \, \in \, \Omega \times \{1, \ldots, C_i\}}{\arg\max} z_i[c].$$

This enables MASS to operate without prior knowledge of the task, adapting its merging and classification on a per-input basis.

## 4 Experiments

**Models and baselines** We run experiments on three CLIP Radford et al. [2021] variants with ViT Dosovitskiy et al. [2021] encoders: ViT-B-32, ViT-B-16, and ViT-L-14. Baselines include training-free methods such as weight averaging, Task Arithmetic Ilharco et al. [2023], and Consensus Merging Wang et al. [2024]. Zero-shot accuracy provides a null reference, while the mean accuracy of individually fine-tuned models serves as the upper bound. We refer to the Appendix for details on our benchmark.

**MoErging results** Tab. 1 shows that MASS sets a new state of the art across all model sizes and task counts, outperforming both classic methods (Task Arithmetic [Ilharco et al., 2023], Consensus TA [Wang et al., 2024]) and newer ones (Iso-C, Iso-CTS [Daniel et al., 2025]).

On the 20-task benchmark, MASS improves absolute accuracy over the fixed `TSV-M` baseline by $+7.4\%$ (`ViT-B-32`), $+4.8\%$ (`ViT-B-16`), and $+2.9\%$ (`ViT-L-14`), while retaining a higher fraction of each model's fine-tuned performance. Gains are largest on smaller backbones, suggesting that routing more effectively mitigates task interference when capacity is limited.

Per-task results (Fig. 2) show consistent improvements, with accuracy retention above $80\%$ for nearly all tasks (even in the 20-task setting) and above $94\%$ in the 8-task benchmark.

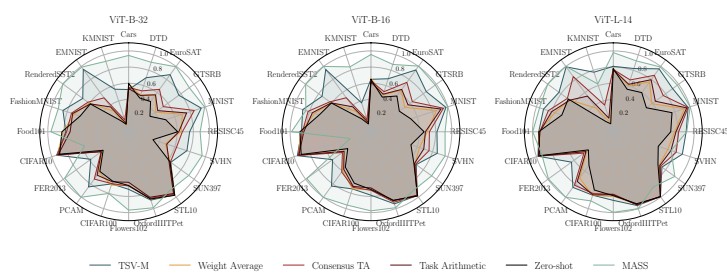

Under the 8-task benchmark on `ViT-B-32`, MASS reaches $97.6\%$ normalized accuracy, surpassing `TwinMerging`'s $95.3\%$ despite not assuming oracle knowledge of the correct head.

Figure 2: Normalized task accuracies for `ViT-B-32`, `ViT-B-16`, and `ViT-L-14` on the 20-task benchmark.

**Choosing a routing layer** We analyze the impact of routing layer choice on task accuracy (Fig. 3 in the Appendix). Both `ViT-B-32` and `ViT-B-16` achieve peak performance at layer 9, with MLP layers slightly outperforming attention layers.

Accuracy, however, varies widely by task, with standard deviations up to $40\%$ across layers. As shown in Fig. 3c, STL10 benefits from earlier layers ($\ell = 3$–$5$), while SUN397 performs best at later ones ($\ell = 9$–$11$). This suggests that optimal routing layers are task-dependent, motivating future work on adaptive selection.

| MASS | ViT-B-32 | | | ViT-B-16 | | |
|---|---|---|---|---|---|---|
| + | 8 tasks | 14 tasks | 20 tasks | 8 tasks | 14 tasks | 20 tasks |
| nn | 94.5 | 91.3 | 91.3 | 93.5 | 91.7 | 86.5 |
| mlp | 98.9 | 98.2 | 96.4 | 98.9 | 98.4 | 95.0 |
| proj$_{PRE}$ | 96.2 | 90.4 | 76.7 | 97.9 | 97.3 | 81.1 |
| proj$_{TSV-M}$ | **97.6** | **95.5** | **92.5** | **98.5** | **97.3** | **91.9** |

Table 2: Average normalized accuracy for different routers.

**Comparison with other routers** Finally, we compare MASS with two common routing strategies, namely:

(i) **Nearest Neighbor (NN)**, which builds a small support set from each task's validation data and assigns a test sample to the nearest embedding; this requires no extra parameters but assumes access to and storage of validation data.

(ii) **MLP router**, which trains an MLP $f_\theta$ on validation embeddings to predict task identity; while accurate, this approach relies on labeled task data, which is often unavailable in practical merging scenarios.

Tab. 2 shows that NN performs well but slightly below MASS, while the MLP achieves the highest accuracy but with limited applicability due to its data requirement. Our projection-based router (`proj`) offers the best balance: starting from `TSV-M` (proj$_{TSV-M}$) outperforms routing from the pretrained backbone (proj$_{PRE}$), as the orthogonal subspaces created by `TSV-M` make residual-based selection effective without any labels or additional training.

## 5 Conclusions

In this paper, we introduced MASS, a merging approach that leverages low-rank task updates while adaptively routing each input to the most relevant subspace. To address the lack of per-task datasets in real-world scenarios, MASS uses a fully data- and training-free projection-based router.

Experiments show that MASS achieves state-of-the-art results, recovering nearly the full accuracy of individual task-specific models at a fraction of their combined storage cost. Future work includes refining the router for finer subspace selection and extending MASS to out-of-distribution scenarios, where task subspaces could be combined on the fly to tackle unseen tasks.

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

# Appendix

We include here additional details on our algorithm, as well as some supplementary results referenced from the main text.

## 5.1 Results

**Benchmark** We evaluate on three collections of tasks, containing 8, 14, and 20 tasks respectively. The latter is the most extensive setup considered in Wang et al. [2024], Gargiulo et al. [2025], Daniel et al. [2025]. The 8-task benchmark, introduced in Ilharco et al. [2023], comprises the following datasets: `Cars` Krause et al. [2013], `DTD` Cimpoi et al. [2014], `EuroSAT` Helber et al. [2019], `GTSRB` Stallkamp et al. [2011], `MNIST` Lecun et al. [1998], `RESISC45` Cheng et al. [2017], `SUN397` Xiao et al. [2016], and `SVHN` Netzer et al. [2011]. Moving to 14 tasks, we add `CIFAR100` Krizhevsky and Hinton [2009], `STL10` Coates et al. [2011], `Flowers102` Nilsback and Zisserman [2008], `OxfordIIITPet` Parkhi et al. [2012], `PCAM` Veeling et al. [2018], and `FER2013` Goodfellow et al. [2013]. The 20-task suite further includes `EMNIST` Cohen et al. [2017], `CIFAR10` Krizhevsky and Hinton [2009], `Food101` Bossard et al. [2014], `FashionMNIST` Xiao et al. [2017], `RenderedSST2` Socher et al. [2013], and `KMNIST` Clanuwat et al. [2018]. We quantify results using both average absolute accuracy and average normalized accuracy.

**Per-layer task accuracy** Results are shown in Fig. 3; we refer to the main text for a discussion.

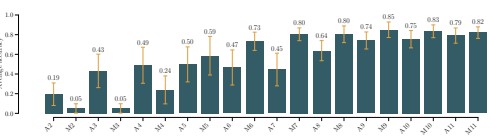

(a) Averaged across all tasks for a `ViT-B-32`.

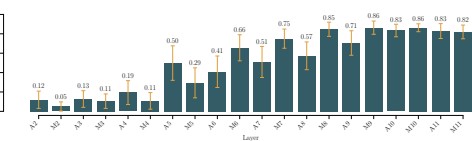

(b) Averaged across all tasks for a `ViT-B-16`.

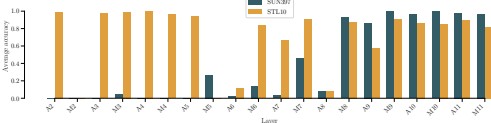

(c) Focusing on `SUN397` and `STL10` for a `ViT-B-32`.

Figure 3: Per-layer task accuracies for `ViT-B-32` on the 20-task benchmark. Layers starting with 'A' indicate attention layers, while those starting with 'M' refer to MLPs.

## 5.2 Algorithms

The key algorithms implementing our method are reported below; we refer to the main text for a discussion.

---

**Algorithm 1** Fixed Merging Step

---

**Require:** Pretrained model weights $\theta_{\text{pre}}$, task-specific updates $\{\Delta_i\}_{i=1}^{T}$, user-specified threshold $\varepsilon$
**Ensure:** Fixed merged model weights $\theta_{\text{MT}}$

1: **Accounting for redundant directions**
2: $\mathcal{M} = \{\}$
3: **for** $i = 1, \ldots, T$ **do**
4:     $\delta_i \leftarrow \text{vec}(\Delta_i)$
5:     **if** $\max_{\{j \in \mathcal{M}\}} \text{sim}(\delta_i, \delta_j) < \varepsilon$ **then**
6:         $\mathcal{M} \leftarrow \mathcal{M} \cup \{i\}$
7:     **end if**
8: **end for**
9: **Merging step using** `TSV-M` Gargiulo et al. [2025] **on the** $\{\Delta_i\}_{i \in \mathcal{M}}$
10: **for** $i \in \mathcal{M}$ **do**
11:     $\Delta_i = U_i \Sigma_i V_i^\top$
12:     $\tilde{U}_i \leftarrow U_{i[:,1:k]}, \tilde{\Sigma}_i \leftarrow \Sigma_{i[1:k,1:k]}, \tilde{V}_i \leftarrow V_{i[:,1:k]}$
13: **end for**
14:     $U \leftarrow [\tilde{U}_1 \,|\, \tilde{U}_2 \,|\, \cdots \,|\, \tilde{U}_T]$
15:     $\Sigma \leftarrow \text{block\_diag}(\tilde{\Sigma}_1, \tilde{\Sigma}_2, \ldots, \tilde{\Sigma}_T)$
16:     $V \leftarrow [\tilde{V}_1 \,|\, \tilde{V}_2 \,|\, \cdots \,|\, \tilde{V}_T]$
17:     $U_\perp \leftarrow \text{orthogonalize}(U)$
18:     $V_\perp \leftarrow \text{orthogonalize}(V)$
19:     $\hat{\Delta} \leftarrow U_\perp \Sigma V_\perp^\top$
20:     $\theta_{\text{MT}} \leftarrow \theta_{\text{pre}} + \alpha \, \hat{\Delta}$
21: **return** $\theta_{\text{MT}}$

---

**Algorithm 2** Adaptive Merging Step

**Require:** Pretrained model weights $\theta_{\text{pre}}$, task-specific updates $\{\Delta_i\}_{i=1}^{T}$, fixed merged model $\theta_{\text{MT}}$,
    top-$k$ parameter $k$, threshold $\eta$, task-specific classification heads $\{h_i\}_{i=1}^{T}$, sample $\mathbf{x}$
**Ensure:** Predicted class $c^*$
  1: $\mathbf{z}_\ell \leftarrow \text{ForwardPass}(\theta_{\text{MT}}, \mathbf{x})$                                       # first pass
  2: **for** $i = 1, \ldots, T$ **do**
  3:     $r_i \leftarrow \|\mathbf{z}_\ell - V_i\,V_i^{\top}\,\mathbf{z}_\ell\|_2$                             # residual as Eq. 4
  4: **end for**
  5: $w \leftarrow \text{softmax}(-r)$
  6: $\Omega \leftarrow \{i \,:\, w_i \geq \eta\}$                                 # Select tasks above threshold
  7: $\Omega \leftarrow \text{TopK}(\Omega, w, k)$                         # Keep only top-$k$ weighted tasks
  8: **Merge selected subspaces**
  9: $\Delta_{\text{ada}} \leftarrow \sum_{i \in \Omega} U_i\,\Sigma_i\,V_i^{\top}$
10: Compute adaptive model: $\theta_{\text{MASS}} \leftarrow \theta_{\text{pre}} + \alpha\,\Delta_{\text{ada}}$
11: **Classification procedure**
12: $\mathbf{z}_{L-1} \leftarrow \text{ForwardPass}(\theta_{\text{MASS}}, \mathbf{x})$                # Compute shared representation
13: $\mathbf{z}_i \leftarrow h_i(\mathbf{z}_{L-1})$                                 # Evaluate each head
14: $(i^\star, c^\star) \leftarrow \underset{(i,c) \in \Omega \times \{1,\ldots,C_i\}}{\arg\max} z_i[c]$            # Highest logit across heads
15: **return** $c^\star$