# OpenReview forum: "MASS: MoErging through Adaptive Subspace Selection"
_NeurIPS.cc/2025/Workshop/UniReps — UniReps2025_

### Official Review · Reviewer_gWxi · 2025-09-06
**MASS: MoErging through Adaptive Subspace Selection**

**Confidence:** 5

**Review:**

## Paper Summary
The paper introduces MASS, a training-free model merging approach that leverages singular-vector-based task updates. MASS dynamically selects relevant subspaces at inference without requiring task data or retraining.

## Quality
The technical quality is strong. The method is well-grounded in prior work on task vectors and task singular vectors. Also, it extends them with adaptive routing inspired by Mixture-of-Experts. The empirical evaluation is thorough and shows clear improvements over baselines. However, efficiency analysis is somewhat limited and not generalized well.

## Clarity
The paper is generally clear and well-written. The structure is easy to follow. Figures and tables help illustrate improvements, and the algorithm pseudocode in the appendix is detailed. Explanations of the projection-based router could be expanded with more details, intuitions.

## Originality
The idea of combining singular-vector-based merging with adaptive inference is novel in the model merging field. The projection-based, data-free router is an original contribution, distinguishing MASS from prior methods.

## Significance
MASS narrows the model merging gap considerably, making it more practical in real-world scenarios where retraining or task data are unavailable. The significance would be stronger if the method were validated on domains beyond image classification.

## Pros
* Novel combination of low-rank task merging and adaptive routing.
* Data-free router avoids reliance on labeled data.
* Strong empirical performance.
* Clear experimental design with comparisons across multiple model scales and task counts.

## Cons
* Evaluation limited to CLIP-based classification; generalizability to other domains is uncertain.
* Inference overhead could be significant in latency-sensitive applications.
* Limited theoretical justification for why residual-based routing works reliably.

## Overall
The paper makes a strong and timely contribution to model merging research. It demonstrates that adaptive, training-free merging can approach the performance of fine-tuned models. Despite some limitations in scope and theoretical analysis, the method is impactful and practical.

**Score:**

4

**Topic Fit:**

3

---

### Official Review · Reviewer_2PZX · 2025-09-14

**Confidence:** 5

**Review:**

## Short Summary:

This submission proposes a training- and data-free model merging method MASS, which leverages MoE-style adaptive router to selects relevant task subspaces for each input based on intermediate feature. With this technique, MASS combines multiple task-specific fine-tuned models weights into a single one for different inputs during inference. MASS is evaluated on several CLIP ViT-based benchmarks with 20 image classification tasks and shows strong performance compared to several static merging baselines. However, the claims of setting a new state-of-the-art are not fully substantiated. The experiments are limited to CLIP image classification. This is a notable limitation, as the paper makes general claims but overlooks the critical and highly relevant vision tasks and other modalities, a major testbed for existing model merging techniques.

---


## Strengths:

**(S1)** MASS is both training and data free at inference time. This is is appealing for practical model merging scenarios like MLLMs and vision foundation models, where only checkpoints are publicly available and task data is missing or is difficult to collect and process.

**(S2)** The adaptive, projection-based router is clearly described and theoretically grounded. It allows input-dependent selection of task subspaces without requiring additional training or access to validation data. This is a significant improvement for multi-task learning over existing input-independent merging approaches.

**(S3)** MASS is benchmarked on 20 image classification tasks and three different sizes of CLIP/ViT models (ViT-B-32, ViT-B-16, ViT-L-14). The empirical results in Table 1, Figure 2, and Page 3 & 4 show consistent improvements in classification performance over prior low-rank and merging baselines.

**(S4)** MASS retains near upper-bound (single task fine-tuned) accuracy, particularly on tasks with smaller backbones where model capacity is limited. This is evidenced by performance exceeding 97% of fine-tuned models in most settings (Table 1).

---

## Weaknesses:

**(W1)** MASS relies on orthogonalization and heuristics whose effects are not theoretically analyzed in depth. For instance, why these particular choices (e.g., block-diagonalization, top-k SVD truncation, cosine similarity thresholds) result in optimal or even stable merging across highly diverse tasks? These obvious questions are not discussed or investigated in the manuscript.

**(W2)** The computational overhead of MASS's two-pass inference (especially latency and memory) is only discussed as 'modest' with relative metrics to a base model, but concrete implications are missing. For large models in real-world deployments (with real-time constraints), a 2x overhead could be prohibitive. This may largely limit the applicability of MASS but is not discussed or empirically examined in the manuscript. IMHO, this could be a risk for this highly efficiency-oriented research direction.

**(W3)** Technical details and guidance for choosing key hyper-parameters in MASS is not fully fleshed out. For example, the routing layer and the thresholds ($k$ and $η$). Other settings like how many singular vectors per task are chosen, exact thresholds for the routing are not detailed or visualized with experiments. This may affect the reproducibility and limit the contribution, since the  model merging field is of high practical demand in real-world deployment.

**(W4)** What the “input” means in lines 32 and 54? Now it is unclear. The task vector? or the task-specific trained model weights? IMHO, this should be precisely clear from the beginning since it is one of the key contributions that correlates with the targeted challenge.

**(W5)** As for the writing, there are too many unnecessary terms defined which negatively impacts the overall readability. For instance, terms like “task subspace” (which exactly is top-k task singular vectors), “task matrix” (which exactly is per-layer task vector), and “lowest-residual tasks” (which is minimal projection error of activations onto the task singular vector spaces) all refer to concepts and terms built on existing model merging or editing studies [1] [2] [3] [4]. These concepts could be expressed using terms in existing model merging studies for better clarity, just as I mentioned in the above parentheses. Yet it seems there are too many unnecessary new terms introduced, which results in significant confusion in reading.

**(W6)** The performance of MASS may not be rigorously examined. This work only compares MASS against several classical baselines like Task Arithmetic and Consensus TA. There are more recent and advanced merging approaches like [5] [6] [7] published in top-tier conferences but are ignored in the manuscript, which makes it difficult to fairly assess the claimed advantages.

**(W7)** The paper title is general and implies broad applicability across diverse tasks and domains. This is also one of the aims of this UniReps workshop. However, the experiments are narrowly focused. All experiments are conducted on CLIP-based image classification tasks with ViT models. There are no evidence, theoretical discussion, or even preliminary results on other domains. Readers seeing this general title would reasonably expect the method to be validated on more than one task and modality. More importantly, model merging is now dominated by interest in LLMs, since techniques to merge LLMs with massive parameters for different skills (e.g., coding, translation, reasoning) are in high demand. As such, MASS sidesteps the most relevant testbed for model merging. IMHO, a claim to a new state-of-the-art is incomplete without addressing this issue. I thus strongly recommend a more accurate title like "MASS: Adaptive Subspace Selection for Vision Model Merging". I also encourage the authors to add a paragraph in Sec. 5 Conclusions to clearly state that the experiments were limited to vision tasks and that future work is required to validate its performance on other modalities.

**(W8) Typos and minor issues.** There is "TwinMerging's 95.3%" in line 119. However, the work "TwinMerging" [8] is not listed as a baseline in Table 1 or mentioned anywhere else (even not in References). This might be a leftover and should be clarified, formally cited or removed. I recommend the authors thoroughly re-examine the manuscript to address similar issues.

---
## Reference

[1] Editing Models with Task Arithmetic, ICLR 2023

[2] Git Re-Basin: Merging Models modulo Permutation Symmetries, ICLR 2023

[3] TIES-Merging: Resolving Interference When Merging Models, NeurIPS 2023

[4] Task Arithmetic in the Tangent Space: Improved Editing of Pre-Trained Models, NeurIPS 2023

[5] AdaMerging: Adaptive Model Merging for Multi-Task Learning, ICLR 2024

[6] CALM: Consensus-Aware Localized Merging for Multi-Task Learning, ICML 2025

[7] Whoever Started the Interference Should End It: Guiding Data-Free Model Merging via Task Vectors, ICML 2025

[8] Twin-Merging: Dynamic Integration of Modular Expertise in Model Merging, NeurIPS 2024


---
## Justification & Message to ACs and the Authors:

Specific questions are already provided in Strengths and Weaknesses. Although there is no rebuttal stage, I still encourage the authors to refine the manuscript based on the comments to further strengthen this work, regardless of its final acceptance status.

I first give a Weak Reject (2) due to the significant concerns about over-claiming, unclear technical descriptions, hyper-parameter details, and presentation clarity. However, the performance of MASS is impressive and might hold practical value in the community. Thus, I believe this submission requires further discussion and I am open to the follow-up exchanges with my fellow reviewers and ACs to reach a consensus for the final recommendation. I hope these comments help ACs understand the basis of my recommendation.

**Score:**

2

**Topic Fit:**

3

---

### Official Review · Reviewer_i14W · 2025-09-15
**Impressive Results, Although Writing Could be Clearer**

**Confidence:** 3

**Review:**

# Summary

The authors perform model merging dynamically per-example. They first use a merged model based on all tasks to evaluate affinity to different tasks. They then use these affinities to decide a subset of tasks which should be used to produce a new merged model, per-sample.

Overall, these are compelling ideas which show clear promise. Even though it comes with some disadvantages, it should serve as valuable inspiration for future methods in model merging and dynamic routing. However, I had some confusion about the method that led to a weak accept rather than a strong one.

# Strengths

- Very impressive result. Exciting!
- Dynamically tuning the merge is an interesting idea, and this version of the idea seems quite successful. (Although it's too bad it needs to be done in a separate forward pass rather than on-the-fly.)

# Weaknesses

- The "Approach" section would benefit from much more detail. This is difficult to fit in a short paper but hopefully could be expanded if this is published as a full paper in the future.
- The selection of routing layer and selection threshold $\eta$ may dampen this method's practical applicability.
- An expanded version of this paper would also benefit from further examination of the computational cost of the method. (Not just 2X inference cost, but cost of merging.)

# Questions / Comments

- What would happen if you used your routing method to simply choose which fine-tuned model to execute, rather than producing a merged model? (The routing itself could still be done with a merged model. But this would avoid having to merge separately for each example.)
- Regarding the 2X storage cost: Doesn't the method require holding weight deltas for all 20 models in memory, and performing merges separately for each individual example, in an unbatched computation?
- Note: it was not explicitly stated, but I assume that merged models comprise all fine-tuned models, not just a subset of them? E.g., for the 20-task benchmark, $\theta_{\mathrm{MT}}$ is the result of merging 20 separately fine-tuned models.
- In Equation (4),
    - is the $V_i$ actually $\tilde{V}_i$? If not, then isn't $V_i V_i^T = I$, the identity matrix?
    - is this $V^{(\ell)}$ or $V^{(\ell+1)}$? I think it must be the latter? In other words, $\mathbf{z}_{\ell}$ is the *input* to this layer? I would advise you to retain the layer index $\ell$ for this case.
    - can you add a sentence explaining the intuitive meaning of $V_i V_i^T \mathbf{z}_{\ell}$? What is this measuring?
- In Section 3, add a reference to Algorithm 2 in the Appendix.
- In Section 3, "Projection-based routing": How is threshold $\eta$ tuned?
- In Section 3, "Adaptive merging and prediction": variables $C_i$ and $\Omega$ were never introduced before now. You also never mentioned retaining multiple task heads, so this whole section is unclear to me. The equation is unclear.
- "MoErging results" section: Here you mention "TwinMerging", but I don't see this mentioned anywhere else, and there is no citation.
- Note: AdaMerging [1] would be a competitive algorithm to add to your evaluations. However, comparing to their 8-task results it does seem like MASS would outperform it. Still, it would come in second, better than the other alternatives you show here. Would also be appropriate to cite.
- "Choosing a router layer" section:
    - I feel that if you do not have room for Figure 3, then the discussion should also be moved to the Appendix.
    - Second, I was surprised when I read this section that you are choosing a single layer to make the decision of which subspaces to merge. I see now that Section 3 says "activations $\mathbf{z}_{\ell}$ at a chosen layer", but this could be emphasized more.
- In Table 2, why did you bold TSV-M when MLP is the best performing? Your method is still attractive due to being data-free, but making it bold is confusing.
    - Also, in this section, it might be helpful to use naming similar to $\theta_{\mathrm{pre}}$ and $\theta_{\mathrm{MT}}$ which you use in Section 3. (maybe proj_MT instead of proj_TSV-M?)

[1] Yang et al. AdaMerging: Adaptive Model Merging for Multi-Task Learning. ICLR 2024.

**Score:**

3

**Topic Fit:**

3

---

### Official Review · Reviewer_G2SR · 2025-09-16
**Review of "MASS: MoErging through Adaptive Subspace Selection"**

**Confidence:** 4

**Review:**

This work presents MASS (MoErging through Adaptive Subspace Selection), a training-free model merging approach that dynamically routes inputs to relevant task subspaces using projection-based residuals. The core idea leverages the low-rank structure of task-specific updates (as observed in prior work like TSV) and introduces a data-free routing mechanism that selects relevant subspaces based on intermediate feature projections. The method achieves impressive results, recovering up to 98% of individually fine-tuned model accuracy across 8, 14, and 20-task benchmarks with only a 2× storage overhead relative to a single pretrained model.

# Strengths
1. Unlike static merging methods, MASS dynamically selects relevant task subspaces per input using a projection-based residual calculation. This eliminates task interference while avoiding the need for task-specific data or additional training.
2. MASS consistently outperforms existing merging techniques across all model sizes (ViT-B-32, ViT-B-16, ViT-L-14) and task counts (8, 14, 20 tasks). On the 20-task benchmark, it improves absolute accuracy over TSV-M by +7.4% (ViT-B-32), +4.8% (ViT-B-16), and +2.9% (ViT-L-14).
3. The method adds only a two-pass inference overhead and ~2× storage cost relative to a single pretrained model, making it highly suitable for deployment scenarios where storage is constrained.
4. The paper evaluates on 20 diverse vision tasks spanning object recognition, scene classification, and fine-grained categorization, with clear comparisons against state-of-the-art baselines

# Weaknesses
1. The entire evaluation is conducted on CLIP-based vision models (ViT variants) with image classification tasks. There is no analysis of how MASS would perform on language models, multimodal models, or other modalities (e.g., audio, time-series). This limits generalizability claims, especially since the workshop focuses on unified representations across modalities.
2. While the projection-based routing is empirically effective, the paper lacks theoretical analysis of why this approach works. For example, how does the residual relate to task similarity? What properties of the task subspaces make them separable via projections? Without this, the method feels more empirical than principled.
3. The paper states MASS adds ~2× storage cost relative to a single pretrained model, but does not compare this to other methods. For example, storing all task-specific models would require 20× storage for 20 tasks, while MASS uses only 2×—but how does this compare to TSV-M or other baselines? A clear comparison would strengthen this claim.

**Score:**

4

**Topic Fit:**

3